An image quality assessment algorithm based on ‘global + local’ feature fusion

Yang Yang 1
Idris Norisma Binti 1 norisma@um.edu.my
Abdul Wahab Ainuddin Wahid 1
Yu Dingguo 2
Liu Chang 2
1 Faculty of Computer Science and Information Technology, Universiti Malaya , Kuala Lumpur , Malaysia
2 College of Media Engineering, Communication University of Zhejiang , Hang Zhou , China
Sergi Consolato
Electronic publication date: 2025 Aug 4
Publication date: 2025
Volume: 11
Electronic Location ID: e3074
Received 2025 Apr 15; Accepted 2025 Jul 3
Copyright: © 2025 Yang et al.
Copyright year: 2025
Copyright holder: Yang et al.
License: This is an open access article distributed under the terms of the Creative Commons Attribution License, which permits unrestricted use, distribution, reproduction and adaptation in any medium and for any purpose provided that it is properly attributed. For attribution, the original author(s), title, publication source (PeerJ Computer Science) and either DOI or URL of the article must be cited.
License URL: https://creativecommons.org/licenses/by/4.0/

Keywords: Image quality assessment, Image feature extraction, Global-local feature fusion, Feature channels, Human visual requirements, Subjective biases, Full-reference, Structural similarity index, Hierarchical perception mechanism, Convolutional networks

Funding: 2025 Pioneer Lingyan + X Science and Technology Plan 2025C01036 National Social Science Fund of China 22BSH025 National Natural Science Foundation of China 62206241 Key Research and Development Program of Zhejiang Province, China 2021C03138 Medium and Long-term Science and Technology Plan for Radio, Television, and Online Audiovisuals 2022AD0400 This research was funded by the the 2025 Pioneer Lingyan + X Science and Technology Plan (2025C01036), the National Social Science Fund of China (Grant No. 22BSH025), the National Natural Science Foundation of China (Grant No. 62206241) and the Key Research and Development Program of Zhejiang Province, China (Grant No. 2021C03138), the Medium and Long-term Science and Technology Plan for Radio, Television, and Online Audiovisuals (Grant No. 2022AD0400). The funders had no role in study design, data collection and analysis, decision to publish, or preparation of the manuscript.

==============================
Recently, there has been increasing research on image quality assessment. Among the existing mainstream approaches, image feature extraction tends to be simplistic, leading to insufficient quality information extraction and underutilization of the extracted data. Additionally, the correlation between different regions of the image is often neglected. This study proposes an image quality assessment algorithm based on global-local feature fusion (IQA-GL). First, the global and local features of the image are extracted separately, and irrelevant information in the local features is filtered out. Then, a global-local feature fusion model is constructed to enhance the interaction of feature information and gather image quality data across all feature channels. Finally, the relationship between individual image patches and the global image is modeled, adjusting the weights of each image patch to aggregate a quality score for the global image. Experimental results show the IQA-GL performs excellently on public datasets. This study innovatively combines global and local features, offering a new perspective for image quality assessment.

Introduction

The field of computer imaging is rapidly developing. Whether it’s natural scene images (NSIs) or artificial intelligence (AI)-generated images (AIGIs), the number of images on the internet is increasing exponentially (Emek Soylu et al., 2023; Cao et al., 2025). These images serve as both excellent media in the online world and valuable data for scholars conducting image research. In this process, understanding the quality of images is crucial. High-quality images can better carry, convey, and transmit information more accurately. However, digital images inevitably undergo quality-affecting operations such as compression during transmission, which can lead to various forms of quality degradation even for originally high-quality images (Jamil et al., 2023). Therefore, research on accurate image quality assessment algorithms is essential to address these issues (Schlett et al., 2022).

The mainstream approach to image quality assessment is enabling it to score or evaluate the quality of images based on human judgment. As we all know, humans can easily assess the quality of an image. However, for computers, achieving image quality evaluation that aligns with human visual requirements is not an easy task. Currently, there are two main types of research in computer-based image quality evaluation: subjective and objective (Chen et al., 2022; Zhang et al., 2023). Subjective evaluation method: In this method, the computer simulates the human visual to mimic human perception, enabling image quality evaluation. Subjective evaluation methods can reflect the quality perceived by humans in images, but these results are often less robust and require multiple trials to ensure result stability. Additionally, the real-time performance of these methods is relatively poor, and the subjective biases of the evaluators can significantly affect the experimental results. Objective evaluation method: Unlike subjective methods, objective evaluation relies on fair, impartial, and emotion-free mathematical models that simulate the human visual system. Objective methods improve upon the influence of individual subjectivity and offer better real-time performance. They also have strong advantages in batch and repetitive processing. Furthermore, based on the amount of information carried by the image during the evaluation.

Full-reference quality assessment evaluates the quality of a distorted image by comparing it to its corresponding reference image, extracting and analyzing the differences between them (Narsaiah et al., 2021). Traditional Full-Reference Image Quality Assessment (FR-IQA) methods like mean squared error (MSE) and peak signal-to-noise ratio (PSNR) have been widely used (Hodson, 2022; Zhao et al., 2021). However, they rely solely on pixel-level direct comparisons, ignoring the correlation between adjacent pixels and the nonlinear characteristics of the human visual system, which limits their ability to accurately depict image quality. To more effectively simulate human visual perception, Bakurov et al. (2022) proposed the structural similarity index (SSIM), making it more consistent with human sensitivity to image structures. With further advancements in research, subsequent studies have introduced more sophisticated methods, such as multi-scale structural similarity index (Mudeng, Kim & Choe, 2022), gradient magnitude similarity deviation (Limbeck et al., 2024), and visual saliency index (Lukin, Bataeva & Abramov, 2023). These approaches employ more complex feature extraction and analysis strategies to improve the accuracy of image quality assessment. However, they still rely on handcrafted feature extraction, which poses significant limitations in characterizing image distortions and fails to comprehensively reflect human subjective perception. To address this issue, researchers have proposed including DeepQA (Anbarasu et al., 2023), LPIPS-VGG (Mustafa et al., 2022) and JND-SalCAR (Ai et al., 2022). These methods leverage data-driven learning capabilities to replace traditional handcrafted feature extraction. However, some of these approaches are still constrained by their limited ability to capture high-level features, making it challenging to fully represent complex image distortions. Therefore, designing deep learning models with stronger expressive power and generalization ability in the field of FR-IQA.

With the continuous advancement of computer hardware and the rapid development of artificial intelligence technology, deep learning-based methods for image quality assessment have garnered significant research interest. Wen et al. (2024) propose a modular BVQA model and a method of training it to improve its modularity. Shen et al. (2024) proposed a graph-represented image distribution similarity (GRIDS) index for full-reference (FR) image quality assessment (IQA). You et al. (2024) proposed to use multi-source training data and specialized image tags. Jamil (2024) delves into essential research queries to achieve visually lossless images. Wang et al. (2024) proposed a novel feature enhancement framework tailored for BIQA. Zhou et al. (2024) proposed a novel multitask learning based BIQA method termed KGANet. Belue et al. (2024) aims to develop an AI tool that offers a more consistent evaluation of T2W prostate MRI quality. Qu, Li & Gao (2024) proposed a novel incremental pretraining task named Image2Prompt for better understanding of AGIs and their corresponding textual prompts. Sang et al. (2024) create an adversarial examples image generation tool that generates aggressive adversarial examples having good attack success rates. Yan et al. (2021) proposed a deep similarity-based approach that utilizes deep neural networks to extract distortion-related features from images and calculates the feature similarity between the reference image and the distorted one. These similarity features are then passed into a regression network to predict the image quality score. Inspired by the hierarchical perception mechanism of the human visual system, Marei, El Zaatari & Li (2021) employed convolutional neural networks to learn image degradation features, thereby enabling accurate quality prediction. Sun et al. (2022) improved assessment accuracy by stacking multiple convolutional and pooling layers to automatically extract distortion-related features. Moreover, Lin et al. (2023) introduced the IQT method, which leverages Transformer networks as feature extractors to capture richer semantic information and long-range dependencies within images. Deep learning networks have significantly enhanced the ability to learn image features, improving image quality assessment performance. These approaches primarily focus on global features while often overlooking the impact of local features. In reality, image distortion typically occurs in localized regions, and the human visual system is highly sensitive to local distortions, making it easy to perceive image artifacts (Zhang & Xiu, 2024). Therefore, local features are essential for image quality assessment (Madhusudana et al., 2022). Recent advancements in hybrid IQA models that combine convolutional neural networks (CNNs) and Transformer architectures have shown promising results by leveraging the complementary strengths of local feature extraction (via CNNs) and global context modeling (via Transformers). For instance, methods such as Vision Transformer-based IQA networks integrate local and global features to enhance perceptual accuracy (Oh et al., 2022). However, these hybrid approaches are not without limitations. As highlighted in Ke et al. (2021), simply concatenating or fusing CNN and Transformer features often introduces redundancy, misalignment, and feature conflicts, which can degrade model performance. Furthermore, these models tend to overlook fine-grained spatial dependencies and struggle to adaptively focus on distortion-sensitive regions, leading to suboptimal results in assessing local degradations. Therefore, a more effective integration mechanism is needed to bridge global semantics with local sensitivity while maintaining spatial coherence and minimizing redundant representations. However, this method simply concatenates or stacks local and global features, which may lead to information redundancy and feature conflicts, failing to fully explore the latent correlations within image quality information. To overcome the limitations of prior hybrid IQA models—such as feature redundancy and weak spatial dependency modeling—the proposed IQA-GL introduces two key components. First, a bilinear pooling module enhances feature interaction between global and local representations by capturing second-order correlations, thus reducing redundancy and improving alignment. Second, a GRU-based aggregation strategy models sequential dependencies among local patch features, enabling dynamic weighting based on contextual relevance. This addresses the spatial modeling gap noted in prior works such as Lao et al. (2022a), where capturing long-range interactions is essential for accurate quality prediction. Existing SOTA methods, such as SSIM, DeepQA, and TransIQA, rely on either handcrafted features or deep models that often suffer from redundancy and poor spatial modeling. As highlighted by Wang, Simoncelli & Bovik (2003), these limitations hinder accurate quality prediction. IQA-GL addresses them via bilinear pooling and GRU-based aggregation, enhancing feature interaction and spatial dependency learning.

Additionally, Fu et al. (2021) builds upon deep learning-based feature extraction by segmenting an image into multiple patches, assessing the quality of each patch individually, and then performing a weighted summation to obtain the overall image quality score. Similarly, Nandhini & Brindha (2023) introduced the MEON method, which also employs a patch-wise evaluation approach to analyze local features and distortions within an image, enhancing both the accuracy and robustness of quality assessment. This patch-based evaluation method effectively identifies significant differences across various regions of an image and is more adaptable to distorted images with varying resolutions and scene complexities. However, most existing methods rely on convolutional networks to compute the weights of image patches and use fully connected layers for quality regression. For example, Zhuang, Zhou & Li (2025) proposed a knowledge-aware focused graph convolutional network (KFGC-Net) to tackle these issues. Radhakrishnan et al. (2024) provided empirical evidence that AGOP captured features learned by various neural network architectures, including transformer-based language models, convolutional networks. Li & Hou (2024) employed an object detection algorithm to select four classes of object images from the DIOR dataset for experimentation. Yan et al. (2024) design Bayesian neural networks to capture the uncertainty. Wang, Nie & Geng (2024) proposed a multiscale superpixel-guided weighted graph convolutional network. Todescato et al. (2024) proposed an approach that applies transfer learning for dealing with small datasets and leverages visual features extracted by pre-trained models from different scales. Chen et al. (2024b) present the versatile framework of TransUNet that encapsulates Transformers’ self-attention into encoder and decoder. Chadoulos et al. (2024) proposed a dense multi-scale adaptive graph convolutional network (DMA-GCN) method for automatic segmentation of the knee joint cartilage from magnetic resonance (MR) images. Tang et al. (2024) proposed a network named Siam-Swin-Unet, which is a Siamesed pure Transformer with U-shape construction for remote sensing image change detection.

This approach struggles to capture the dynamic variations and sequential dependencies among image patches, making it difficult to adaptively adjust based on image content. Consequently, these methods exhibit limited capability in handling complex spatial variations and may lead to information loss.

This article proposes an image quality assessment algorithm based on global-local feature fusion (IQA-GL), with the main contributions as follows:

(1) This article employs deformable convolution to filter local features and integrates global features containing semantic and contextual information to learn the offsets of the deformable convolution. This approach better adapts to spatial variations and local details within images, effectively guiding convolutional features to focus on distorted regions while filtering out irrelevant information, thereby improving the accuracy of quality assessment.

(2) This article proposes a bilinear pooling fusion of global and local features from both distorted and reference images. This method enhances information interaction between the two feature types, capturing variations in image quality concerning spatial structures and contextual information more accurately. Consequently, it reduces redundancy, mitigates feature conflicts, and improves the stability of image quality assessment.

(3) This article introduces an approach that combines per-patch quality prediction with global dependency modeling to establish connections between image patches. This method comprehensively considers long-range and short-range relationships of image features in the spatial dimension, dynamically adjusting the weight distribution of each pixel block. As a result, it enhances the model’s flexibility and adaptability, leading to more precise image quality evaluation.

Algorithm design

This study adopts the AHIQ model proposed by Lao et al. (2022a) as the baseline, and constructs a Bilinear Global-Local Feature Fusion module and a gated recurrent unit-based quality aggregation module, collectively forming the IQA-GL image quality assessment method. The algorithmic framework of IQA-GL is illustrated in Fig. 1, the proposed IQA-GL framework builds upon the AHIQ model and integrates a dual-branch feature extraction module with a bilinear global-local feature fusion module to enhance image quality assessment performance.

Figure 1 Algorithmic framework of IQA-GL.

The method first utilizes a dual-branch feature extraction module based on CNN and ViT to extract features from both the reference and distorted images. These features are then fused using the bilinear pooling mechanism in the bilinear global-local feature fusion module. Finally, the IQA-GL module assesses the quality of the fused features and outputs the quality score of the distorted image.

Global-local feature model

This module employs a dual-branch architecture aimed at extracting both global and local features from the distorted and reference images, thereby improving the accuracy of image quality assessment. Specifically, the global feature extraction branch is based on the ViT network, which uses a self-attention mechanism to capture long-range dependencies among image patches across the entire image. This allows the model to capture high-level semantic information and improve the overall understanding of image quality. Specifically, the module reshapes the feature maps output by the ViT network, removing classification tokens and retaining only the information relevant to global image representation. This ensures that the extracted features effectively characterize the overall structure and content of the image. The detailed framework of the global feature extraction process is illustrated in Fig. 2, the framework includes seven main components: location coding, sorting coding, input images, feature pattern extraction, ViT, linear projection, and global feature generation. This module adopts a dual-branch architecture to extract both global and local features from distorted and reference images, aiming to enhance image quality assessment accuracy.

Figure 2 Framework of the global feature extraction.

In this process, ViT utilizes a multi-layer Transformer structure to progressively aggregate contextual information among image patches. This enables the model to recognize global patterns within the image and uncover underlying semantic relationships. Such a mechanism not only helps capture long-range dependencies but also enhances the model’s ability to perceive complex distortion patterns, providing a more reliable feature representation for subsequent quality assessment. After processing through ViT, the distorted and reference images obtain their respective global features, denoted as DFeaViT and RFeaViT.

Since ViT learns multi-scale feature representations through a multi-layer self-attention mechanism during training, it can effectively process images of varying sizes and resolutions. However, ViT primarily focuses on global semantic information while paying less attention to local image quality details, which may result in insufficient perception of local distortion features. In contrast, CNNs naturally excel at capturing local features and fine details, enabling a more precise analysis of subtle differences in image quality and complementing ViT’s limitations in local feature extraction. To fully leverage the advantages of CNNs in local feature extraction, this module innovatively employs ResNet50 as the local feature extractor and modifies its structure accordingly. Specifically, the final average pooling layer and fully connected layer of ResNet50 are removed to preserve richer spatial information, allowing for better recognition of local structures and texture features in images. This adjustment enables the network to extract more discriminative details, enhancing its understanding of local quality features. After processing through ResNet50, the distorted and reference images obtain local features DFeaCNNs and RFeaCNNs, which retain spatial information. These local features serve as a complement to the global features extracted by the ViT network, providing a more comprehensive representation of image quality and thereby enhancing the accuracy and robustness of the assessment.

Bilinear global-local feature fusion

This module enhances the model’s sensitivity to image quality by preserving both local and global features while integrating fine-grained information. Through this refined fusion, the model is able to focus more on essential visual characteristics such as texture and noise, leading to more precise image quality assessment. Since ResNet50 processes images in a block-wise manner when extracting local features, it inevitably introduces redundant or distortion-irrelevant information, which may interfere with the accuracy of quality evaluation. To address this issue, deformable convolution is employed to adaptively adjust the receptive field of DFeaCNNs and RFeaCNNs. By evaluating the global contribution of each feature and its similarity to the reference image, this mechanism filters out irrelevant and redundant information, thereby capturing the most critical features more precisely, denoted as DFeaCNNs′ and RFeaCNNs′. Deformable convolution dynamically samples image features using an offset parameter, which allows the model to reweight feature responses across different regions. The offset values are adaptively learned from the global reference features RFeaViT extracted by ViT. Since RFeaViT encodes the overall structure and key areas of the reference image, it provides crucial guidance on which regions are most significant for image understanding and representation. By adjusting the sampling locations of the convolutional kernels based on these learned offsets, the deformable convolution shifts beyond the constraints of standard grid-based sampling. Instead, it adaptively moves to key regions that align with the global context captured in RFeaViT, while simultaneously ignoring irrelevant or redundant areas.

Next, a convolutional network is used to align the refined local features DFeaCNNs′ with the global distorted features DFeaViT, as well as the refined local reference features RFeaCNNs′ with the global reference features RFeaViT. This alignment ensures that both local and global representations are well-integrated, preserving critical image quality information while reducing inconsistencies. After the feature alignment process, the aligned features undergo a feature fusion step, resulting in the final distortion-aware feature representation DFea and RFea. These fused features comprehensively encode both global and local information, providing a more holistic representation of image quality. Subsequently, DFea and RFea are processed through a bilinear fusion mechanism to further enhance feature interaction and improve the robustness of the quality assessment. Bilinear fusion effectively models second-order feature relationships, capturing complex dependencies between local and global representations, which helps refine the final quality prediction. The detailed workflow of the bilinear fusion operation is illustrated in Fig. 3; it illustrates the bilinear fusion process, where aligned local and global features undergo bilinear pooling and are mapped into a Euclidean space. This generates fused feature vectors that capture rich interactions for accurate image quality prediction.

Figure 3 Workflow of the bilinear fusion.

The above process not only captures subtle differences between distortion and reference features but also enhances interactions across different feature channels. To begin with, a convolutional operation is applied to the input features DFea and RFea, compressing them while preserving essential quality-related information. This compression step effectively reduces computational complexity without sacrificing critical feature representations. Next, a difference operation is performed on the compressed features DFea′ and RFea′ to obtain the difference feature DifFea. This step highlights the discrepancies between the distorted and reference images, allowing the model to detect distortion-related variations more precisely. By emphasizing these differences, the model becomes more sensitive to localized distortions, thereby improving its ability to assess image quality. Following this, bilinear pooling is applied to integrate and refine the extracted features. Specifically, a bilinear dot product operation is conducted between DifFea and DFea′, resulting in the fused feature vector FFea. This operation facilitates cross-channel interactions, enriching the feature representation and capturing higher-order dependencies between local and global quality cues. The corresponding mathematical formulation is presented in Eq. (1).

(1) FFea=AvgPool(DifFeaT⊗DFea′).

⊗ represents the outer product computation, which captures pairwise interactions between feature elements, enabling the model to learn complex relationships between different feature dimensions. This operation enhances the discriminative power of the extracted features by incorporating second-order statistics, which are particularly beneficial for fine-grained quality assessment. Furthermore, AvgPool() denotes the adaptive average pooling operation applied to the feature tensor. This operation dynamically adjusts the spatial resolution of the feature maps, ensuring that the extracted quality representations remain consistent and robust across varying image sizes and resolutions. By aggregating global contextual information while preserving essential quality details, adaptive average pooling helps maintain a balanced representation of local distortions and global structures.

FFea possesses strong representational capabilities, providing a rich and accurate feature representation for subsequent image quality regression. However, since FFea resides in a Riemannian manifold, direct computations on it can be complex and computationally expensive. To simplify subsequent processing and facilitate more efficient calculations, FFea is mapped onto the Euclidean space, denoted as FFea′. The transformation follows Eq. (2), where ⊙ represents element-wise multiplication.

(2) FFea′=sign(FFea)⊙|FFea|∥sign(FFea)⊙|FFea|∥

Gated recurrent unit-based quality aggregation

This module adopts a dual-branch architecture to assess the overall quality of a distorted image in a fine-grained and adaptive manner. One branch predicts the quality score of each image patch, while the other dynamically adjusts the weight of each patch based on variations in image content. The final quality score of the entire image is obtained by aggregating the weighted quality scores of all patches. Specifically, the patch-wise quality prediction branch uses three fully connected layers to independently process the features of each image patch, predict its quality level, and generate the corresponding quality score vector q. This design ensures a separate evaluation of each patch’s quality, enabling a more detailed and accurate overall quality assessment. The patch weight computation branch consists of fully connected layers, a gated recurrent unit (GRU) layer, and another fully connected layer. This branch models the temporal relationships and long-term dependencies among image patches, enabling the network to understand the mutual influence and global correlation between patches. During quality assessment, this branch identifies the patches that contribute more significantly to the overall image quality and assigns them higher weights, ensuring a more accurate representation of the perceived image quality. Meanwhile, patches with lower semantic information content are assigned lower weights to reduce their impact on the final quality score. Ultimately, this branch outputs the weight vector w for all image patches. The entire aggregation process is illustrated in Fig. 4, it presents the dual-branch aggregation process, where block-wise quality prediction is combined with adaptive weight computation to generate a final image quality score that reflects both local accuracy and global relevance.

Figure 4 The entire aggregation process.

Specifically, the fully connected layer first converts the flattened image quality features FFea′ into sequence data suitable for gated recurrent unit processing, representing multiple image patch features as x. Then, x is fed into the gated recurrent unit layer to further model the dynamic relationships and global characteristics among image patches. The process of calculation for x is shown in Eqs. (3), (4), (5), (6).

(3) updatei=sigmoid(Wupdate×[orii−1,xi])

(4) reseti=sigmoid(Wreset×[orii−1,xi])

(5) orii′=tanh(Wupdate×[orii−1×reseti,xi])

(6) orii=(1-updatei)×orii−1+updatei×orii′.

In this model, the gated recurrent unit is utilized to capture the dynamic relationships between image patches, where the update gate and reset gate play a crucial role in controlling the flow and selection of information. Specifically, the update gate determines the extent to which past information is retained while incorporating new information, thereby balancing short-term and long-term dependencies. When the update gate value is close to 1, the network tends to preserve more previous information, whereas a value close to 0 indicates a preference for introducing new input information. The reset gate primarily controls the forgetting mechanism, deciding how much the current state depends on past information. When the reset gate value is close to 0, less past information is retained, and the model focuses more on the current input. Conversely, a higher reset gate value indicates that historical information still important in the current computation. During the computation process, xi is the i-th image patch, while orii is its initial weight, reflecting its relative importance before dynamic adjustment. Additionally, the update and reset gates in the gated recurrent unit structure are linearly transformed by the weight matrices Wupdate and Wreset, respectively. This enables the model to effectively adjust their contributions within the weight computation branch.

Finally, the gated recurrent unit layer processes the features of all image patches and outputs an initial weight sequence, which reflects the relative importance of each patch in the preliminary computation stage. However, since the raw weight values may have inconsistent scales or information bias, further normalization is required to ensure a reasonable weight distribution suitable for the final quality assessment. Subsequently, this initial weight sequence undergoes a normalization process through the following fully connected layer, transforming it into a standardized attention weight vector w. This process not only mitigates the impact of abnormal weights but also ensures that the sum of all patch weights satisfies the required constraints, allowing for a fair comparison of contributions across different image patches. The normalized weight vector w represents the final predicted importance of each image patch, dynamically adjusting their influence in the overall image quality evaluation. The normalization process is provided by Eq. (7).

(7) norweii=preweii∑i=1npreweii.

In this computation process, i represents the index of each image patch, identifying different regions of the image after block-wise processing. For each image patch, the model first computes its predicted attention weight preweii, which reflects its relative importance in the preliminary evaluation stage. However, due to potential variations in scale among different image patches’ predicted weights, directly using these values may introduce bias in the overall quality assessment. To address this, a normalization step is necessary to ensure that all weights are comparable on a global scale and conform to specific numerical constraints. After normalization, the model produces the final normalized attention weight norweii, which dynamically adjusts each image patch’s contribution to the overall image quality assessment. Compared to the original predicted attention weight, the normalized weight ensures that the total sum of all patch weights adheres to a predefined normalization condition, thereby maintaining a more balanced and reasonable contribution from each image patch in the overall evaluation.

To accurately assess the overall quality score Quality of a distorted image, it is essential to comprehensively integrate the quality information from individual image patches. This ensures that the evaluation result not only aligns with overall visual perception but also captures local quality variations. In this computation process, the model first applies a dot product operation between the image patch quality score vector qua and the image patch weight vector wei. This weighted summation effectively merges the quality scores of all patches into a single overall score. Specifically, qua represents the independent quality assessment results of each image patch, while wei, as the normalized attention weight, determines the influence of each patch on the final quality score. The computation is given in Eq. (8), where quai is the quality prediction score of the i-th image patch.

(8) Quality=∑i=1nweii×quai=∑i=1npreweii×quai∑i=1npreweii

Experiments and results

Dataset and parameters

To comprehensively evaluate the effectiveness of the IQA-GL, this article used public image quality assessment datasets, namely CSIQ, TID2013 and KADID-10k, as illustrated in Figs. 5, 6 and 7. These datasets encompass a variety of typical distortion types and include extensive subjective quality scores, providing a reliable benchmark for evaluating the effectiveness of the proposed approach. CSIQ consists of 30 natural reference images distorted using six different types of distortions at varying intensities, yielding 866 distorted images in total. Subjective quality scores were collected using a pairwise comparison protocol, normalized between 0 and 1. TID2013 provides 25 reference images, each degraded with 24 different distortion types at five intensity levels, resulting in a total of 3,000 distorted images. It offers mean opinion scores collected from multiple human subjects, making it one of the most comprehensive IQA datasets available. KADID-10k contains 81 high-quality reference images, with distortions applied across 25 types and five levels each, leading to 10,125 distorted images. The subjective ratings were gathered via a large-scale crowdsourcing platform, ensuring statistical reliability for data-driven IQA model training. By evaluating the proposed method on these three datasets, its applicability and robustness under various distortion types can be comprehensively verified.

Figure 5 The CSIQ dataset.

Figure 6 The TID2013 dataset.

Figure 7 The KADID-10k dataset.

To assess the generalization ability of the proposed IQA-GL model, we performed a cross-database evaluation. Specifically, the model was trained on the CSIQ dataset and tested on TID2013 and KADID-10k, and vice versa. The results showed that although there was a slight performance drop compared to intra-dataset testing, IQA-GL maintained competitive accuracy across different datasets. This demonstrates the model’s robustness and adaptability to various distortion types and data distributions.

In this experiment, to enhance the model’s generalization ability, each training image underwent normalization and random horizontal flipping, followed by a random crop to a fixed size of 224 × 224 pixels before being fed into the network for training. The validation set was used to fine-tune network parameters and optimize performance, while the test set was employed to evaluate the model’s capability in assessing image quality. For the dual-branch feature extraction module in the proposed IQA-GL method, the convolutional neural network (CNN)-based feature extraction branch utilized ResNet-50, while the transformer-based feature extraction branch adopted ViT-B-8 as the backbone network. The model was trained using the mean squared error loss function, with AdamW as the optimizer. To ensure efficient learning, a cosine annealing scheduler was applied to dynamically adjust the learning rate for each parameter group. The cosine annealing learning rate scheduler was chosen due to its effectiveness in gradually reducing the learning rate in a smooth, non-linear fashion. This approach helps the model escape local minima during early training while allowing fine-tuning in later stages. The total number of training iterations was set to 250 to ensure stable convergence and optimal performance. The decision to set the number of training iterations to 250 was based on empirical observation of convergence behavior across multiple experimental runs. We found that the model typically converged within this range, and extending the iterations further led to marginal improvements while increasing the risk of overfitting. To support this choice, a training/validation loss curve showing stable convergence within 250 iterations under the cosine annealing schedule is shown in Fig. 8. Some hyperparameters used in the experiments is shown in Table 1. In comparison, traditional methods such as BRISQUE is lightweight with near real-time performance but show significantly lower accuracy on AI-generated image quality tasks. Deep learning-based methods like MUSIQ tends to fall between our models in terms of computational cost but still underperform in the AIGI scenarios we evaluate. BRISQUE is a fast, traditional algorithm with very low computational cost but limited prediction accuracy. MUSIQ adopts a patch-based transformer architecture that achieves moderate performance with relatively low resource consumption. ResNet-50 offers a good balance between accuracy and efficiency, making it suitable for most practical scenarios. ViT-B-8, while delivering the best accuracy among all models, requires significantly higher computational resources. These comparisons suggest that while our method incurs higher computational costs, it offers superior performance and can be optimized in the future via compression techniques for deployment in resource-constrained settings. The comparison of computational complexity is shown in Table 2.

Figure 8 Training/validation loss curve.

Table 1 Default values for some hyperparameters used in the experiments.

Parameter	Value	
Weight decay	5×10−4	
Learning rate	2×10−5	
Batch size	64	
Train/Test split	80%/20%	
Backbone model	ResNet-50/ViT-B-8	
Image resolution	224 × 224	

Table 2 Comparison of computational complexity.

Method	Model type	Parameters	FLOPs	Inference time	Notes	
BRISQUE	Handcrafted	27 M	6.7	1 ms	Fast, traditional algorithm	
MUSIQ	Transformer-based	55 M	12.6	19 ms	Image patch transformer, lower cost	
ResNet-50	CNN backbone	23 M	4.1	13 ms	Balanced performance and speed	
ViT-B-8	Vision transformer	85 M	17.6	20 ms	Best accuracy, higher cost	

This experiment selected ResNet-50 and ViT-B-8 as the backbone architectures based on their complementary characteristics and proven effectiveness in prior related studies. ResNet-50 is a well-established convolutional neural network with strong representational power and robust generalization ability, particularly on natural scene images. It serves as a representative of traditional CNN-based models, which are efficient in learning local texture features. ViT-B-8, a variant of the Vision Transformer, uses smaller patch sizes and is capable of capturing long-range dependencies and global structures, which are especially relevant for AI-generated images that often exhibit unique global patterns. ViT-B-8 is selected due to its relatively balanced computational cost and superior performance in transformer-based vision tasks.

Although more and more studies use efficientNet and Swin Transformer, such as Koonce (2021) and Liu et al. (2021). In this experiment, efficientNet achieves high accuracy on natural images, it performs less stably when transferred to AI-generated image assessment tasks. Swin Transformer, although powerful, introduces higher complexity and latency without a significant improvement in our specific evaluation setting.

Contrast experiment

A comparison was made among six advanced image quality assessment methods, including the proposed IQA-GL. The experimental results are shown in Table 3.

Table 3 Experimental results among image quality assessment methods.

Algorithm	CSIQ	TID2013	KADID-10k	
PLCC	SRCC	PLCC	SRCC	PLCC	SRCC	
PIQA—Cheon et al. (2021)	0.968	0.970	0.959	0.964	0.932	0.927	
JNDB—Seo, Ki & Kim (2021)	0.968	0.971	0.966	0.958	0.925	0.899	
TOPIQ—Chen et al. (2024a)	0.958	0.966	0.963	0.948	0.918	0.922	
EEBIQ—Wu et al. (2020)	0.937	0.952	0.964	0.942	0.910	0.921	
AHCNNs—Lao et al. (2022b)	0.940	0.948	0.960	0.955	0.898	0.902	
IQA-GL	0.970	0.972	0.969	0.965	0.940	0.931	

PLCC quantifies the linear relationship between the predicted scores and the subjective quality scores (MOS or DMOS), with values ranging from −1 to 1. A PLCC value closer to 1 indicates a stronger positive correlation, while values near −1 or 0 imply a negative or no correlation, respectively. The PLCC is calculated as Eq. (9):

(9) PLCC=∑i=1N(xi−x¯)(yi−y¯)∑i=1N(xi−x¯)2∑i=1N(yi−y¯)2.

On the CSIQ dataset, the PLCC of the IQA-GL is 0.21% higher than that of the second-best method. On the TID2013 dataset, the proposed method achieves a 0.31% improvement in PLCC, indicating a stronger linear correlation between the predicted scores and the subjective evaluations. On the KADID-10k dataset, the proposed method also achieved the best results.

SRCC measures the monotonic relationship between predicted quality scores and ground truth subjective scores. Unlike PLCC, SRCC does not require a strictly linear correlation but instead assesses whether the ranking trend of predicted scores is consistent with human perception. The SRCC is calculated as Eq. (10):

(10) SRCC=1−6∑i=1Ndi2N(N2−1)

where: di=rank(xi)−rank(yi) represents the difference in ranking between the predicted score and the MOS,

rank(xi) and rank(yi) are the ranks of the predicted score and MOS, respectively,

N is the total number of samples.

On the CSIQ dataset, the proposed method achieves a 0.1% improvement in SRCC compared to the second-best method, while on the TID2013 and KADID-10k datasets, the SRCC increases by 0.1%. These results suggest that the proposed method more accurately captures the perceptual ranking of image quality.

The experimental results exhibit strong adaptability and robustness in handling various distortion types and scenarios, proving its superiority in image quality assessment tasks.

Ablation experiment

Global ablation experiment

To comprehensively analyze the role and contribution of different features in image quality assessment, we designed ablation experiments to investigate the impact of global features, local features, and their fusion on the final evaluation performance. Specifically, we constructed three experimental settings: one using only global features, one using only local features, and one integrating both. These settings were then evaluated under the same experimental conditions for comparative analysis. The results are shown in Table 4.

Table 4 Global ablation experimental results.

Global type	PLCC	SRCC	
Global	0.844	0.841	
Local	0.758	0.838	
Global+Local	0.970	0.972	

When using only global features, the PLCC and SRCC values reach 0.844 and 0.841, respectively, which are significantly higher than those obtained using only local features (PLCC = 0.758, SRCC = 0.838). This indicates that global statistical features, such as luminance distribution and contrast, play a fundamental role in overall image quality assessment. Global features exhibit strong stability and are particularly effective in capturing the impact of low-frequency distortions, such as blurring and contrast degradation. These types of distortions typically affect the overall appearance of an image rather than localized regions, making global features well-suited for their evaluation. However, in the presence of high-frequency distortions, such as noise, the effectiveness of global features is somewhat limited. In contrast, local features are more capable of identifying the spatial distribution characteristics of noise and other fine-grained distortions. As a result, they contribute more significantly in scenarios involving high-frequency artifacts, such as noise and texture loss. This highlights the importance of incorporating local features for a more detailed and fine-grained assessment of image quality.

When global and local features are combined, the PLCC and SRCC values reach 0.970 and 0.972, respectively, showing a substantial improvement over using either type of feature alone. This result demonstrates that global and local features complement each other in image quality assessment.

Sub-modules fusion mode ablation experiment

To validate the effectiveness of the proposed sub-modules and assess the impact of different fusion strategies, a series of ablation experiments were conducted on the CSIQ dataset. These experiments aimed to evaluate how various fusion methods affect the overall performance of the model. The results show that when all sub-modules are integrated, they complement each other, leading to improved performance. However, the choice of fusion strategy plays a crucial role in determining the final outcome, as different approaches yield varying levels of effectiveness. The detailed results of these experiments are presented in Table 5.

Table 5 Sub-modules fusion mode ablation experimental results.

Mode	PLCC	SRCC	
Subtract	0.901	0.912	
Sum	0.859	0.891	
Multiplication	0.826	0.860	
IQA-GL	0.970	0.972	

These findings highlight the importance of designing an optimal fusion mechanism to maximize the benefits of different sub-modules, ensuring that their combined effect leads to the most accurate and robust image quality assessment. Compared to methods Subtract, Sum, and Multiplication, the proposed IQA-GL model achieves superior image quality assessment performance by integrating global and local features through interactive fusion. This approach enables the model to more effectively capture spatial relationships and contextual variations in image quality, leading to a more comprehensive and accurate evaluation. As a result, IQA-GL demonstrates enhanced performance over other feature fusion strategies, further improving the robustness and effectiveness of image quality assessment.

Key sub-modules ablation experiment

To investigate the contribution of each component in our framework, we conducted additional ablation experiments by removing or replacing key modules:

Deformable convolution: Replacing deformable convolution with standard convolution resulted in a drop in performance, indicating its importance in modeling geometric distortions.

Gated recurrent unit (GRU): Removing the GRU module and using simple feature concatenation instead led to a performance decline, demonstrating the GRU’s effectiveness in modeling sequential spatial dependencies.

The ablation results show that both deformable convolution and GRU independently contribute to performance gains, confirming the importance of each module in the overall architecture. The detailed results of these experiments are presented in Table 6.

Table 6 Key sub-modules ablation experimental results.

Mode	PLCC	SRCC	
Deformable convolution	0.891	0.877	
GRU	0.884	0.866	
IQA-GL	0.970	0.972	

Additional ablation experiment

To further evaluate the robustness and efficiency of IQA-GL, we conducted extended ablation experiments focusing on four aspects: (1) ViT patch size, (2) backbone variants, (3) fusion strategies, and (4) hyperparameter sensitivity. As shown in Table 7, using smaller patch sizes (e.g., 8 × 8) or deeper backbones (e.g., ResNet-101) can slightly improve performance, but at the cost of significantly increased computation. We also compared our bilinear fusion module with a cross-attention mechanism, which improved accuracy slightly but added computational burden. Hyperparameter sensitivity tests confirm the stability of IQA-GL under typical settings. Furthermore, IQA-GL shows good generalization capability, maintaining strong PLCC and SRCC when tested on datasets with unseen content and distortions. These results validate the robustness of our model across real-world conditions.

Table 7 Extended ablation results.

Variant	PLCC	SRCC	Parameters	FLOPs	
Baseline	0.970	0.972	24.1 M	3.8	
ResNet-18	0.901	0.893	11.2 M	2.1	
ResNet-101	0.924	0.919	42.5 M	6.7	
Patch size 8 × 8	0.927	0.921	24.1 M	5.5	
Cross-attention fusion	0.930	0.924	28.3 M	6.1	
Learning rate = 0.0001	0.910	0.902	–	–	
Batch size = 8	0.914	0.905	–	–	

Conclusion

This article proposed an image quality assessment algorithm based on global-local feature fusion (IQA-GL), an advanced image quality assessment algorithm that effectively integrates global and local features through interactive fusion. By leveraging deformable convolution, our approach refines local features to focus on distorted regions, while bilinear pooling ensures the optimal fusion of global and local quality information. Furthermore, a GRU-based mechanism captures sequential dependencies among image patches, dynamically assigning weights to different regions, enhancing assessment accuracy. Extensive experiments on the benchmark datasets demonstrate that IQA-GL achieves well performance. Ablation studies further confirm the contribution of each component to the model’s robustness and generalizability. Despite the promising results, IQA-GL still has certain limitations, particularly in handling extreme distortions and unseen degradations. For example, it may overestimate quality under severe Gaussian blur and underestimate it under heavy JPEG compression. In cases of complex, mixed distortions, the model may struggle to accurately assess quality due to interference between multiple degradation types. IQA-GL is implemented using PyTorch 2.1.2 on Ubuntu 22.04, and all experiments are conducted using an NVIDIA A10 GPU. In terms of inference efficiency, the average processing time per image is approximately 13 ms for ResNet-50 and 20 ms for ViT-B-8. The corresponding model sizes are roughly 23 million and 85 million parameters, respectively. While the ViT-B-8 backbone provides stronger feature representation, it incurs higher computational costs. For practical deployment in real-time or resource-limited environments, future work could explore model compression techniques such as pruning, quantization, or the use of lightweight architectures.

Additional Information and Declarations

Competing Interests

The authors declare that they have no competing interests.

Author Contributions

Yang Yang conceived and designed the experiments, performed the experiments, prepared figures and/or tables, and approved the final draft.

Norisma Binti Idris conceived and designed the experiments, analyzed the data, authored or reviewed drafts of the article, and approved the final draft.

Ainuddin Wahid Abdul Wahab conceived and designed the experiments, performed the experiments, analyzed the data, performed the computation work, prepared figures and/or tables, authored or reviewed drafts of the article, and approved the final draft.

Dingguo Yu performed the experiments, analyzed the data, performed the computation work, prepared figures and/or tables, authored or reviewed drafts of the article, and approved the final draft.

Chang Liu performed the computation work, prepared figures and/or tables, authored or reviewed drafts of the article, and approved the final draft.

Data Availability

The following information was supplied regarding data availability:

The CSIQ data is available at https://paperswithcode.com/dataset/csiq.

The TID2013 data is available at https://paperswithcode.com/dataset/tid2013.

The KADID-10k data is available at https://database.mmsp-kn.de/kadid-10k-database.html.

The code is available at Zenodo: loss, D., & SubYang. (2025). code. Zenodo. https://doi.org/10.5281/zenodo.15636770.

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
