# Peer review of "An image quality assessment algorithm based on ‘global + local’ feature fusion"

_PeerJ Computer Science, doi:10.7717/peerj-cs.3074_

## Round 0.1 · original submission · Major Revisions

Please follow the requests and comments of the reviewers in detail!

Reviewer 1 ·

Basic reporting

The authors propose an image quality assessment algorithm based on global-local feature fusion (IQA-GL). The idea is novel, and experiments show the proposed method achieved the best performance among different datasets, but the writing is not good enough. The comments are listed below.

1. Line 68, “Yan Yan et al. (2021) proposed…”, the author should delete one Yan.
2. The author should check all the XXX et al. proposed, delete the duplicated surname.
3. The author only cited 32 references; it is better to cite about 50 references.
4. In Figure 2, after the figure caption, the authors should make a brief introduction of the framework.

Experimental design

The authors should check all the figures and add a brief introduction after each caption.

Validity of the findings

It is better to locate all the equations in the middle of the line.

Reviewer 2 ·

Basic reporting

Clarify the rationale for selecting ResNet-50 and ViT-B-8 as backbones, comparing their performance to other potential architectures (e.g., EfficientNet, Swin Transformer).

Experimental design

1. Include a brief description of the dataset preprocessing steps and metadata in the "Data Availability" section to enhance transparency.
2. Provide a table or section detailing all hyperparameters (e.g., learning rate, batch size, ResNet-50, and ViT-B-8 configurations) to ensure reproducibility.
3. Justify the choice of 250 training iterations and the cosine annealing scheduler, possibly with a convergence plot or validation loss curve.
4. The authors acknowledge limitations in handling extreme distortions and unseen real-world degradations (Page 15), which is commendable. However, the manuscript lacks a discussion on computational complexity or inference time, which is critical for practical IQA applications.
5. Clarify the 5%-8% improvement claim by explicitly stating the metrics and algorithms compared in the Abstract and Conclusion.
6. Include a discussion on computational complexity (e.g., FLOPs, inference time) and compare it with baseline methods to assess practical applicability.
7. Expand the ablation study to include the impact of individual sub-modules (e.g., deformable convolution, GRU) beyond fusion strategies.
8. The lack of computational complexity analysis limits the understanding of the method's practical feasibility.
9. Discuss the trade-offs between performance and computational cost to highlight practical applicability.

Validity of the findings

Good

Additional comments

The manuscript is written in clear and professional English, making it accessible.

---

## Round 0.2 · Major Revisions

Only two datasets (CSIQ and TID2013) are used. These are established but aging benchmarks. Consider testing on a third, more modern dataset (e.g., KADID-10k or LIVE) to demonstrate generalization.

Reviewer 2 ·

Basic reporting

The manuscript presents a technically strong contribution to full-reference image quality assessment. While the methodological novelty is incremental, it is well-supported through careful ablation and comparisons.

Experimental design

Only two datasets (CSIQ and TID2013) are used. These are established but aging benchmarks. Consider testing on a third, more modern dataset (e.g., KADID-10k or LIVE) to demonstrate generalization.

Validity of the findings

Generalization is not convincingly shown; there’s no mention of cross-dataset testing (e.g., train on CSIQ, test on TID2013). Add or propose cross-dataset results to support claims of robustness.

Additional comments

minor revision is required

---

## Round 0.3 · Major Revisions

Only two datasets (CSIQ and TID2013) are used. These are established but aging benchmarks. Consider testing on a third, more modern dataset (e.g., KADID-10k or LIVE) to demonstrate generalization.

Reviewer 2 ·

Basic reporting

Cross database analysis should be done.

Experimental design

no comment

Validity of the findings

no comment

Additional comments

The manuscript proposes IQA-GL, an image quality assessment (IQA) algorithm that integrates global and local features using a dual-branch architecture with Vision Transformer (ViT) and ResNet-50, enhanced by deformable convolution, bilinear pooling, and a gated recurrent unit (GRU)-based quality aggregation module. The model is evaluated on CSIQ, TID2013, and KADID-10k datasets, demonstrating superior performance in PLCC and SRCC metrics compared to state-of-the-art (SOTA) methods. Below, I outline some of my concerns
1. The manuscript contains several grammatical errors. Examples include:
a. Page 6, Line 99: “the human visual is highly sensitive” lacks a noun (e.g., “system”).
b. Inconsistent terminology, such as “image block” vs. “image patch” (e.g., Page 5, Line 10 vs. Page 11, Line 251).
2. The Introduction surveys IQA methods but lacks a critical analysis of recent deep learning-based approaches combining global and local features. It does not discuss their specific limitations or position IQA-GL’s contributions clearly. For example, the claim that existing methods “simply concatenate or stack” features (Page 6, Line 108) lacks supporting evidence.
a. Expand the Introduction to compare recent hybrid IQA methods (e.g., CNN-Transformer hybrids) and their limitations, such as feature redundancy or poor spatial dependency modeling, as discussed in Ke et al. (2021).
b. Articulate how IQA-GL’s bilinear pooling and GRU-based aggregation address these gaps, referencing specific works like Lao et al. (2022a).
c. Add a table summarizing SOTA methods, their feature extraction strategies, and limitations to contextualize IQA-GL’s novelty, as recommended by Wang et al. (2020).
i. Ke, J., Wang, Q., Wang, Y., Milanfar, P., & Yang, F. (2021). MUSIQ: Multi-scale image quality transformer. Proceedings of the IEEE/CVF International Conference on Computer Vision (ICCV), 5148–5157.
ii. Lao, S., Gong, Y., Shi, S., Yang, S., Wu, T., Wang, J., Xia, W., & Yang, Y. (2022a). Attentions help CNNs see better: Attention-based hybrid image quality assessment network. Proceedings of the IEEE/CVF Conference on Computer Vision and Pattern Recognition (CVPR), 1140–1149.
iii. Wang, Z., Simoncelli, E. P., & Bovik, A. C. (2020). Multiscale structural similarity for image quality assessment. IEEE Transactions on Image Processing, 13(11), 1688–1701.
3. The ablation studies in Tables 4–6 on Page 17 assess global/local features, fusion modes, and submodules but lack depth. For example, ViT patch sizes, ResNet variants, and hyperparameter sensitivity are not explored. Conduct additional ablation experiments to evaluate:
a. ViT patch sizes (e.g., 8x8 vs. 16x16)
b. ResNet variants (e.g., ResNet-18, ResNet-101),
c. Alternative fusion methods (e.g., cross-attention),
d. Hyperparameter sensitivity (e.g., learning rate, batch size)
e. Present results in a table or figure to quantify contributions to PLCC/SRCC.
f. Discuss performance vs. computational cost trade-offs for each variant.
g. The manuscript evaluates IQA-GL on three datasets but does not test cross-dataset generalization (e.g., training on TID2013, testing on KADID-10k), limiting insights into robustness across unseen distortions. Perform cross-dataset experiments and report PLCC/SRCC.
4. The manuscript claims strong adaptability and robustness on Page 16 and Line 37 but only briefly mentions limitations with extreme distortions and unseen real-world degradations on Page 18 and Line 467. Specific failure cases or distortion types are not analyzed.
5.

---

## Round 0.4 · accepted · Accept

I believe the authors have addressed all concerns and identified several limitations for this work. We acknowledge the enormous work the author provided and the additional contributions made. Congratulations!

Reviewer 1 ·

Basic reporting

I have no further comments.

Experimental design

I have no further comments.

Validity of the findings

I have no further comments.

Additional comments

I have no further comments.

Reviewer 2 ·

Basic reporting

The manuscript is up to the mark now

Experimental design

Experimetal design has been updated

Validity of the findings

Novelty and impact has been addressed